# Improved Detection Criteria for Detecting Drug-Drug Interaction Signals Using the Proportional Reporting Ratio

**DOI:** 10.3390/ph14010004

**Published:** 2020-12-23

**Authors:** Yoshihiro Noguchi, Keisuke Aoyama, Satoaki Kubo, Tomoya Tachi, Hitomi Teramachi

**Affiliations:** 1Laboratory of Clinical Pharmacy, Gifu Pharmaceutical University, 1-25-4, Daigakunishi, Gifu-shi, Gifu 501-1196, Japan; 175003@gifu-pu.ac.jp (K.A.); 175050@gifu-pu.ac.jp (S.K.); tachi@gifu-pu.ac.jp (T.T.); 2Laboratory of Community Health Pharmacy, Gifu Pharmaceutical University, Daigakunishi, Gifu-shi, Gifu 501-1196, Japan

**Keywords:** spontaneous reporting systems, drug-drug interaction, proportional reporting ratio, combination risk ratio, concomitant signal score

## Abstract

There is a current demand for “safety signal” screening, not only for single drugs but also for drug-drug interactions. The detection of drug-drug interaction signals using the proportional reporting ratio (*PRR*) has been reported, such as through using the combination risk ratio (*CRR*). However, the *CRR* does not consider the overlap between the lower limit of the 95% confidence interval of the *PRR* of concomitant-use drugs and the upper limit of the 95% confidence interval of the *PRR* of single drugs. In this study, we proposed the concomitant signal score (*CSS*), with the improved detection criteria, to overcome the issues associated with the *CRR*. “Hypothetical” true data were generated through a combination of signals detected using three detection algorithms. The signal detection accuracy of the analytical model under investigation was verified using machine learning indicators. The *CSS* presented improved signal detection when the number of reports was ≥3, with respect to the following metrics: accuracy (*CRR*: 0.752 → *CSS*: 0.817), Youden’s index (*CRR*: 0.555 → *CSS*: 0.661), and *F*-measure (*CRR*: 0.780 → *CSS*: 0.820). The proposed model significantly improved the accuracy of signal detection for drug-drug interactions using the *PRR*.

## 1. Introduction

Pre-marketing randomized clinical trials typically focus on establishing the safety and efficacy of a single drug rather than investigating drug-drug interactions; therefore, patients who use drugs other than the one under investigation are usually excluded. However, unlike pre-marketing trials, it is common to use multiple drugs for treatment post-marketing. Therefore, attention should be paid not only to the adverse events caused by a given drug, but also to those arising as a result of interactions between two or more drugs. In some reports, the proportion of adverse events caused by drug-drug interactions was estimated to be approximately 30% of the unexpected adverse events [1]. The use of a spontaneous reporting system is believed to be beneficial for the early detection of drug-induced adverse events post-marketing. Spontaneous reporting systems do not include the number of drug users; therefore the incidence of adverse events cannot be calculated, and instead, unknown adverse events are searched for using safety signals [2].

Several detection algorithms [3,4,5,6] have been developed based on disproportionality analysis, for example, the proportional reporting ratio (*PRR*) [3] is often used as an algorithm for single-drug-induced adverse events. In addition to these algorithms, several signal detection algorithms for drug-drug interactions have been reported [7,8,9].

Susuta et al. proposed the combination risk ratio (*CRR*) as a signal detection algorithm for drug-drug interactions [10]. As suggested by these authors, the *CRR* is calculated by dividing the estimated *PRR* points of concomitant-use *drug D*_1_ and *drug D*_2_ (=*PRR _drug D_*_1 ∩ *drug D*2_) by the estimated *PRR* points of *drug D*_1_ or *drug D*_2_ (*PRR _drug D_*_1_ or *PRR _drug D_*_2_), as shown in Table 1 and Equations (1)–(3) [10].

However, to calculate the *PRR _drug D_*_1 ∩ *drug D*2_ and the *χ*^2^
*_drug D_*_1 ∩ *drug D*2_, replace it as follows: *N*_11_ = *n*_111_, *N*_00_ = *n*_000_ + *n*_010_ + *n*_100_, *N*_10_ = *n*_110_, *N*_01_ = *n*_001_ + *n*_011_ + *n*_101_, *N*_1+_ = *n*_11+_, *N*_+1_ = *n*_++1_, *N*_0+_ = *n*_00+_ + *n*_01+_ + *n*_10+_, *N*_+0_ = *n*_++0_. Identically, to calculate (1) the *PRR _drug D_*_1_ and the *χ*^2^
*_drug D_*_1_, (2) the *PRR _drug D_*_2_ and the *χ*^2^
*_drug D_*_2_, replace it as follows:

(1) *N*_11_ = *n*_111_ + *n*_101_, *N*_00_ = *n*_000_ + *n*_010_, *N*_10_ = *n*_110_ + *n*_100_, *N*_01_ = *n*_001_ + *n*_011_, *N*_1+_ = *n*_11+_ + *n*_10+_, *N*_+1_ = *n*_++1_, *N*_0+_ = *n*_00+_ + *n*_01+_, *N*_+0_ = *n*_++0_,

(2) *drug D*_2_: *N*_11_ = *n*_111_ + *n*_011_, *N*_00_ = *n*_000_ + *n*_100_, *N*_10_ = *n*_110_ + *n*_010_, *N*_01_ = *n*_001_ + *n*_101_, *N*_1+_ = *n*_11+_ + *n*_01+_, *N*_+1_ = *n*_++1_, *N*_0+_ = *n*_00+_ + *n*_10+_, *N*_+0_ = *n*_++0_.
(1)$$Combination\;risk\;ratio\;\left(CRR\right)=\frac{PRR_{drug\;D_{1}\cap \;drug\;D_{2}}}{\max\left(PRR_{drug\;D_{1}},\;PRR_{drug\;D_{2}}\right)}\;$$
(2)$$PRR=\frac{\left(N_{11}/N_{1+}\right)}{\left(N_{01}/N_{0+}\right)}\;$$
(3)$$\chi^{2}=\frac{n_{+++}\times {\left(\left|N_{11}\times N_{00}-N_{10}\times N_{01}\right|-n_{+++}/2\right)}^{2}}{N_{1+}\times N_{+1}\times N_{0+}\times N_{+0}}\;$$

When (1) *n*_111_ ≥ 3, (2) *PRR _drug D_*_1 ∩ *drug D*2_ > 2, (3) *χ*^2^
*_drug D_*_1 ∩ *drug D*2_ > 4, (4) *CRR* > 2, this was signal of drug-drug interaction.

However, the number of adverse events reported during concomitant use is generally lower than that of single-drug-induced adverse events, and the 95% confidence interval (95%CI) tends to be wider for *PRR _drug D_*_1 ∩ *drug D*2_. In other words, as shown in Figure 1, when the individual 95%CIs are considered, it is possible that the lower limit of the 95%CI of *PRR _drug D_*_1 ∩ *drug D*2_ (=*PRR*_025_
*_drug D_*_1 ∩ *drug D*2_) overlaps with the upper limit of the 95%CI of *PRR _drug D_*_1_ or *PRR _drug D_*_2_ (=*PRR*_025_
*_drug D_*_1_ or *PRR*_025_
*_drug D_*_2_).

If such an overlap occurs, a risk signal may not be detected for concomitant use. Therefore, with reference to the interaction signal score (*INTSS*) [11], we proposed the concomitant signal score (*CSS*) (Figure 2) and verified its applicability for improving the detection criteria for the *CRR* proposed by Susuta et al [10].

## 2. Results

### 2.1. Model Evaluation Using Receiver Operating Characteristic (ROC) and Precision-Recall (PR) Curves and the Area Under the Curve (AUC)

The receiver operating characteristic (ROC) curve and precision-recall (PR) curve of the *CSS* are shown in Figure 3. The area under the curve (AUC) score was 0.86 for the ROC (cutoff value: 1.041, Youden’s index (max): 0.632, and *F*-measure: 0.638) and 0.570 for PR (cutoff value: 1.271, Youden’s index: 0.623, and *F*-measure (max): 0.645).

### 2.2. Model Evaluation Using Machine Learning Indicators

In all cases, 739 pairs were detected using the *CRR* (true positive (*TP*): 473, false positive (*FP*): 266, true negative (*TN*): 2735, and false negative (*FN*): 450). In contrast, the *CSS* detected 1862 signal pairs (*TP*: 882, *FP*: 980, *TN*: 2021, and *FN*: 41).

The numbers of *TP*, *FP*, *TN*, and *FN* signal pairs detected for *n*_111_ ≥ 3 are shown in Table 2. A total of 739 pairs were detected using the *CRR* (*TP*: 473, *FP*: 266, *TN*: 335, and *FN*: 1), and 621 pairs were detected using the *CSS* (*TP*: 449, *FP*: 172, *TN*: 429, and *FN*: 25).

The *CSS* confirmed an improvement in signal detection with respect to the following metrics: (1) Youden’s index (*CRR*: 0.424 → *CSS*: 0.629), (2) *F*-measure (*CRR*: 0.569 → *CSS*: 0.633), (3) Recall (sensitivity) (*CRR*: 0.512 → *CSS*: 0.956), and (4) negative predictive value (NPV) (*CRR*: 0.859 → *CSS*: 0.874), in comparison with *CRR*.

Furthermore, for *n*_111_ ≥ 3, the *CSS* confirmed an improvement in signal detection with respect to the following metrics: (1) Accuracy (*CRR*: 0.752 → *CSS*: 0.817), (2) Youden’s index (*CRR*: 0.555 → *CSS*: 0.661), (3) *F*-measure (*CRR*: 0.780 → *CSS*: 0.820) (4) Specificity (*CRR*: 0.557 → *CSS*: 0.714), and (5) Precision (positive predictive value; PPV) (*CRR*: 0.640 → *CSS*: 0.723). The metrics of the Ω shrinkage measure were as follows: (1) Accuracy (0.880), (2) Youden’s index (0.775), (3) *F*-measure (0.875), (4) Specificity (0.824), (5) Recall (sensitivity) (0.951), (6) Precision (PPV) (0.810), and (7) NPV (0.956) (Table 3).

### 2.3. Cohen’s Kappa Coefficient

The similarity metrics between the *CSS* and the Ω shrinkage measure were *κ* (95%CI): 0.330 (0.314–0.345), *P*_positive_: 0.505, and *P*_negative_: 0.759 in all case. On the other hand, for *n*_111_ ≥ 3, those values were *κ* (95%CI): 0.729 (0.708–0750), *P*_positive_: 0.880, and *P*_negative_: 0.849.

## 3. Discussion

We evaluated the setting criteria and accuracy of the *CSS*, a newly proposed analytical model to overcome potential issues with the *CRR*. In this study, 3924 pairs of *drug D*_1_–*drug D*_2_—Stevens–Johnson syndrome (SJS) in the Japanese Adverse Drug Event Report (JADER) database were evaluated. Unknown adverse event data do not exist, therefore no “real” true data for adverse events were available. Therefore, to verify the accuracy of *CSS*, we had to prepare “hypothetical” true data pertaining to adverse events. These “hypothetical” true data were also used to validate the subset analysis for detecting drug-drug interaction signals [12].

The ROC curve was generated to determine the cutoff value of *CSS*. The highest value for Youden’s index was 0.632, and the cutoff value was 1.041 (*F*-measure: 0.645). Furthermore, the PR curve must be considered when evaluating imbalanced data such as those used in the present study. The *F*-measure was the highest for a cutoff value of 1.271 (*F*-measure: 0.645, Youden’s index: 0.623). Considering the detection criteria, including the results presented here, the criterion (*CSS* > 1) proposed by us would be preferable.

The highest number of signals was detected using the *CSS*, with 1862 pairs (47.5% of the total combinations, accuracy: 0.740, Youden’s index: 0.629, and *F*-measure: 0.633), followed by the *CRR,* with 739 pairs (18.8% of the total combinations, accuracy: 0.817, Youden’s index: 0.424, *F*-measure: 0.569), in all cases. One reason for this difference in the number of detections is that, unlike the *CSS*, the *CRR* cannot detect combinations for *n*_111_ < 3. Therefore, we also investigated the difference in the number of detections for *n*_111_ ≥ 3.

As shown in Table 2 and Table 3, 621 signal pairs were detected using the *CSS* (57.8% of the total combinations, accuracy: 0.817, Youden’s index: 0.661, and *F*-measure: 0.820), whereas 739 pairs were detected using the *CRR* (68.7% of the total combinations, accuracy: 0.752, Youden’s index: 0.556, and *F*-measure: 0.780). These results demonstrate the significantly improved signal detection accuracy of the newly proposed *CSS* in comparison with that of the *CRR*. However, the *CSS* exhibited slightly lower accuracy for detecting drug-drug interaction signals in comparison with the Ω shrinkage measure.

These results suggest that the *CSS* might be a more suitable method for detecting drug-drug interaction signals using *PRR* instead of the *CRR*.

Our previous studies [13] have shown that the Ω shrinkage measure is the most conservative signal detection method. Therefore, in this study, we also investigated the similarity between the *CSS* and the Ω shrinkage measure.

The similarity metrics between the *CSS* and the Ω shrinkage measure were *κ* (95%CI): 0.330 (0.314–0.345), *P*_positive_: 0.505, and *P*_negative_: 0.759, whereas those between the *CRR* and Ω shrinkage measure were *κ*: 0.718 (0.703–0.733), *P*_positive_: 0.771, and *P*_negative_: 0.948 [13] in all case. As mentioned earlier, although the *CRR* cannot detect combinations for *n*_111_ < 3, the *CSS* can. It is considered that this difference in detection criteria affected the similarity.

Indeed, the *CSS* is more similar to the Ω shrinkage measure than the *CRR* in *n*_111_ ≥ 3; the Ω shrinkage measure and *CSS* was *κ* (95%CI): 0.729 (0.708–0750), *P*_positive_: 0.880, and *P*_negative_: 0.849, whereas the *CRR* and Ω shrinkage measure were *κ*: 0.621 (0.597–0.646), *P*_positive_: 0.850 and *P*_negative_: 0.763 [13]. Unlike the *CRR*, the *CSS* has the advantage of being able to detect combinations for *n*_111_ < 3, but if a more conservative signal detection is needed, consider adding “*n*_111_ ≥ 3” to the criteria.

However, this study, like our previous study [12], has three limitations:(1)The true data used in this study consist of a statistics-based *drug D*_1_–*drug D*_2_ adverse event (SJS) combination rather than a pharmacology-based combination. Unfortunately, data on unknown adverse events do not exist; thus, it was necessary to use “hypothetical” true data instead of “real” true data for validation. Therefore, we used a combination of signals detected using the three algorithms as “hypothetical” true data for detecting drug-drug interaction signals.(2)It is important to compare detection trends using all adverse events recorded in the validation datasets. However, numerous combinations of *drug D*_1_–*drug D*_2_ adverse events are expected, and a study design including all such combinations is not practical. Therefore, SJS was the target adverse event in this study. Although this adverse event has been used previously in other comparative studies by our group [12,13] and other researchers [10,14], it is possible that different performance characteristics are obtained when different reference sets are used.(3)Differences in the approach adopted by regulatory authorities may result in differences in the tendency to register adverse events in the spontaneous reporting system. For example, JADER has long not accepted reports from patients, whereas the Food and Drug Administration Adverse Events Reporting System (FAERS) includes reports from non-medical professionals. It is unclear how the differences in registration tendencies would affect the results of this study [12]. However, as verified by Caster et al. [15], the statistical impact of differences in the number of cases enrolled in the spontaneous reporting system on this study might be small.

## 4. Materials and Methods

### 4.1. Data Sources

The validation dataset was created from the Japanese Adverse Drug Event Report database (JADER), using data from the first quarter of 2004 to the fourth quarter of 2015. It can be accessed directly here: (http://www.info.pmda.go.jp/fukusayoudb/CsvDownload.jsp) (in Japanese only).

### 4.2. Definitions of Adverse Drug Events

The drugs targeted for the survey were all registered and classified as “suspect drugs” in verification data set. The adverse event targeted for this study was Stevens–Johnson syndrome (SJS) using the preferred term (PT) in the Medical Dictionary for Regulatory Activities Japanese version (MedDRA/J).

### 4.3. “Hypothetical” True Data of Adverse Events for Comparative Verification

There are no “real” true data for unknown adverse events. It was considered that only known adverse events should not be used as “real” true data for validation, because detection algorithms require the power to detect unknown adverse events. Therefore, “hypothetical” true data used in the previous study [12] were also used in this study.

“Hypothetical” true data used the combination of signals detected by three algorithms (the additive model [16], the multiplicative model [16], and the Chi-squared statistical test [17]).

### 4.4. Statistical Models and Criteria

#### 4.4.1. Previous Model (combination risk ratio) and criteria proposed by Susuta et al.

This model proposed by Susuta [10] is calculated using Figure 1 as described and Equations (1)–(3) in the introduction section. When (1) *n*_111_ ≥ 3, (2) *PRR drug D*_1_ ∩ *drug D*_2_ > 2, (3) *χ*^2^
*drug D*_1_ ∩ *drug D*_2_ > 4, (4) combination risk ratio (*CRR*) > 2, this was signal of drug-drug interaction.

#### 4.4.2. New Model (Concomitant Signal Score) and Criteria

Two new models and criteria were considered. The lower 95%CI for the *PRR* (*PRR*_025_) and the upper 95%CI for the *PRR* (*PRR*_975_) was calculated using Figure 1 and Equations (2) and (4).
(4)$$PRR\left(95\%\mathrm{CI}\right)=\;e^{ln\;\left(PRR\right)\pm 1.96\sqrt{\frac{1}{N_{11}}-\frac{1}{N_{1+}}+\frac{1}{N_{01}}-\frac{1}{N_{0+}}}}$$

We proposed the *CSS* (Equation (5)) as a new model. The newly proposed model is the ratio of *PRR*_025_
*_drug D_*_1 ∩ *drug D*2_ and *PRR*_975_
*_drug D_*_1_ (or *PRR*_975_
*_drug D_*_2_). This principle is similar to the interaction signal score (*INTSS*) [16].

In this study, we considered (1) *PRR*_025_
*_drug D_*_1 *∩ drug D*2_ > 1, (2) *CSS* > 1 as the newly proposed criteria.
(5)$$Concomitant\;signal\;score\;\left(CSS\right)=\frac{PRR_{\;025\;drug\;D_{1}\cap \;drug\;D_{2}}}{\max\left(PRR_{\;975\;drug\;D_{1}},\;PRR_{\;975\;drug\;D_{2}}\right)}\;$$

#### 4.4.3. Model (Ω Shrinkage Measure) and Criteria to be Compared.

In this study, we selected the Ω shrinkage measure [18] as the model for comparison. The Ω shrinkage measure shows the most conservative detection trends of the five algorithms based on frequency statistical model for drug-drug interactions in our previous study [13].

The calculation is shown in Equations (6) and (7). However, *E*_111_ is the expected value of adverse event (AEs) caused by the combination of two drugs. Ω_025_ > 0 is used as a threshold to screen for signals under the combination of two drugs
(6)$$\Omega =log_{2}\frac{n_{111}+0.5}{E_{111}+0.5}\;$$
(7)$$\Omega_{025}=\Omega -\frac{\phi \left(0.975\right)}{log\left(2\right)\sqrt{n_{111}}}\;$$

### 4.5. Evaluation of Models for Detection

#### 4.5.1. Model Evaluation Using the ROC and PR Curves and AUC

The ROC curve is normally used to judge model performance. To properly analyze imbalanced data, which include a higher proportion of negative data than positive data, evaluation using the PR curve [19,20] in addition to the ROC curve is necessary. Therefore, the PR curve was also used in this study.

#### 4.5.2. Using Evaluations of Classification in Machine Learning

The evaluation indicators that we have set were Accuracy (Equation (8)), Youden’s index (Equation (9)), *F*-measure (Equation (10)), Specificity, Recall (=Sensitivity), Precision (=Positive predictive value; PPV), and negative predictive value (NPV).
(8)$$\mathrm{Accuracy}=\frac{TP+TN}{TP+FP+TN+FN}\;$$
(9)$${\mathrm{Youden}}^{\prime }s\;\mathrm{index}=\;\mathrm{Sensitivity}+\;\mathrm{Specificity}-1\;$$
(10)$$F-\mathrm{measure}=\frac{2\times \mathrm{Recall}\times \mathrm{Precision}}{\mathrm{Recall}+\mathrm{Precision}}\;$$

#### 4.5.3. Cohen’s Kappa Coefficient

The commonality of the signals detected by each statistical model was evaluated using Cohen’s kappa coefficient (*κ*), proportionate agreement for positive rating (*P*_positive_), and proportionate agreement for negative rating (*P*_negative_), as reported in previous studies [12,13,14]. In this study, we investigated the similarities with the Ω shrinkage measure for the previous/newly proposed analysis (*CRR*/*CSS*).

### 4.6. Analysis Software

The analysis software in this study used Visual Mining Studio (NTT DATA Mathematical Systems Inc., Shinjuku-ku, Tokyo, Japan) version 8.4, Microsoft Excel 2019 (Microsoft Corp., Redmond, WA, USA), and R version 4.0.0 (R Core Team, https://www.R-project.org/) with PRROC package version 1.3.1.

## 5. Conclusions

Polypharmacy has become a contemporary social problem, which has necessitated safety signal screening, not only for single drugs but also for drug-drug interactions. A convenient method is sought because most methods for detecting drug interaction signals involve complicated calculations.

The *CRR* proposed by Susuta et al. is based on *PRR*, which not only facilitates the detection of drug-drug interaction signals, but also makes it easy to understand the fluctuations in drug-drug interactions due to a single drug in terms of signal intensity [10]. However, unlike the *INTSS*, *CRR* does not consider the overlap between the lower limit of the 95%CI of *PRR _drug D_*_1 ∩ *drug D*2_ (=*PRR*_025_
*_drug D_*_1 ∩ *drug D*2_) and the upper limit of the 95%CI of *PRR _drug D_*_1_ or *PRR _drug D_*_2_ (=*PRR*_025_
*_drug D_*_1_ or *PRR*_025_
*_drug D_*_2_).

In this study, we proposed a *CSS* with an improved detection criteria, with reference to the *INTSS*, to overcome the issues associated with the *CRR*. Our proposed model significantly improved the accuracy of signal detection for drug-drug interactions using the *PRR*.

## Figures and Tables

**Figure 1 pharmaceuticals-14-00004-f001:**
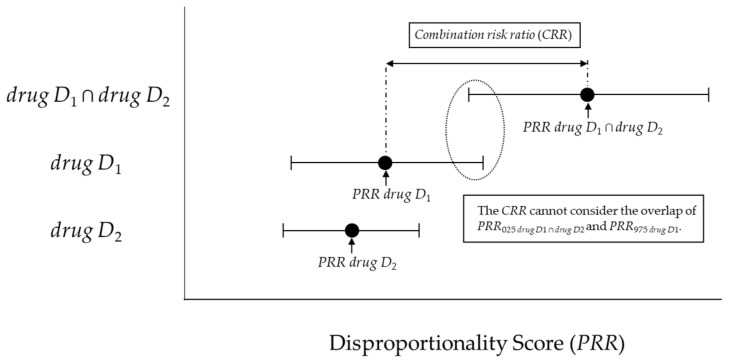
The association between the combination risk ratio (*CRR*) and disproportionality score.

**Figure 2 pharmaceuticals-14-00004-f002:**
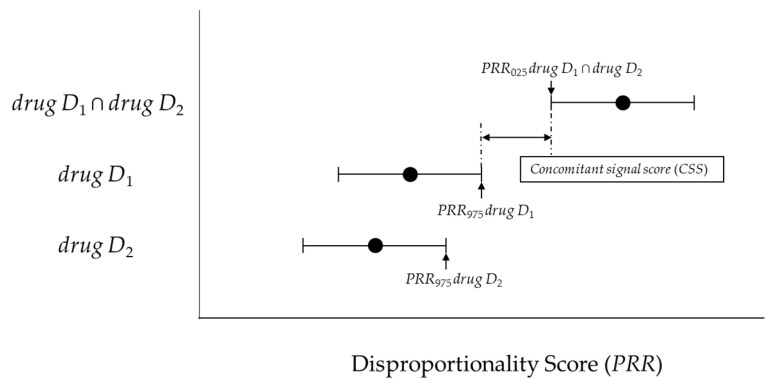
The association between the concomitant signal score (*CSS*) and disproportionality score.

**Figure 3 pharmaceuticals-14-00004-f003:**
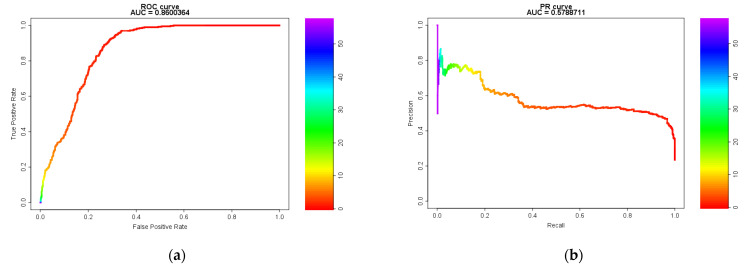
Model Evaluation Using Receiver Operating Characteristic (ROC) and Precision-Recall (PR) Curves and the Area Under the Curve (AUC). (**a**) the receiver operator characteristic curve; (**b**) the precision-recall curve.

**Table 1 pharmaceuticals-14-00004-t001:** The 4 × 2 contingency table for signal detection: AE: adverse event; *n*: the number of reports.

	Target AE	Other AEs	Total
**Concomitant use of *drug D*_1_ and *drug D*_2_**	*n* _111_	*n* _110_	*n* _11+_
**only *drug D*_1_**	*n* _101_	*n* _100_	*n* _10+_
**only *drug D*_2_**	*n* _011_	*n* _010_	*n* _01+_
**Neither *drug D*_1_ or *drug D*_2_**	*n* _001_	*n* _000_	*n* _00+_
**Total**	*n* _++1_	*n* _++0_	*n* _+++_

**Table 2 pharmaceuticals-14-00004-t002:** The numbers of true positive, false positive, true negative, and false negative signal pairs for *n*_111_ ≥ 3.

Analytical Model	*TP*	*FP*	*TN*	*FN*
Combination risk ratio (*CRR*)	473	266	335	1
Concomitant signal score (*CSS*)	449	172	429	25
Ω shrinkage measure	451	106	495	23

*TP*: true positive, *FP*: false positive, *TN*: true negative, *FN*: false negative.

**Table 3 pharmaceuticals-14-00004-t003:** Evaluation of the drug-drug interaction signals detected for *n*_111_ ≥ 3.

Analytical Model	Accuracy	Youden’s Index	*F*-Measure	Specificity	Recall (Sensitivity)	Precision (PPV)	NPV
Combination risk ratio (*CRR*)	0.752	0.555	0.780	0.557	0.998	0.640	0.997
Concomitant signal score (*CSS*)	0.817	0.661	0.820	0.714	0.947	0.723	0.944
Ω shrinkage measure	0.880	0.775	0.875	0.824	0.951	0.810	0.956

PPV: positive predictive value, NPV: negative predictive value.

## Data Availability

Please refer to suggested Data Availability Statements in section “MDPI Research Data Policies” at https://www.mdpi.com/ethics.

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
