# Peer review of "Improved Detection Criteria for Detecting Drug-Drug Interaction Signals Using the Proportional Reporting Ratio"

_pharmaceuticals, 2020, doi:10.3390/ph14010004_

Round 1

Reviewer 1 Report

The authors propose an elegant new method of signal detection for drug interactions (Concomitant Signal Score), improving it as compared with the “traditional” Combination Risk Ratio method. The CSS method improves the drug interaction signal detection for a number of cases ≥3 with a good accuracy compared to CRR, even if still low as compared with the Ω shrinkage measure method. The manuscript is relevant and well-written. There seems to be a typographic error in the body of the text, though: the upper limit for the PRR for single drugs is given as PRR025 instead of PRR975, as mentioned in the figure legends.

Author Response

Reviewer #1

We thank the reviewer for the constructive opinion.

We made all necessary changes to the referee and highlighted them in the manuscript (yellow).

Page 6 Line 211

and PRR025 drug D1 (or PRR025 drug D2). → and PRR975 drug D1 (or PRR975 drug D2).

Reviewer 2 Report

The authors present a well-written manuscript with a new detection criteria for detecting drug-drug interaction trhough the proportional reporting ratio algorithm. I did not find any area where major revision was required. 

Author Response

Reviewer #2

We thank you for the proper review.